# Omics and Multi-Omics in IBD: No Integration, No Breakthroughs

**DOI:** 10.3390/ijms241914912

**Published:** 2023-10-05

**Authors:** Claudio Fiocchi

**Affiliations:** 1Department of Inflammation & Immunity, Lerner Research Institute, Cleveland, OH 44195, USA; fiocchc@ccf.org; 2Department of Gastroenterology, Hepatology and Nutrition, Digestive Disease and Surgery Institute, Cleveland Clinic, Cleveland, OH 44195, USA

**Keywords:** inflammatory bowel disease, ulcerative colitis, Crohn’s disease, omics, multi-omics, systems biology, network medicine, artificial intelligence

## Abstract

The recent advent of sophisticated technologies like sequencing and mass spectroscopy platforms combined with artificial intelligence-powered analytic tools has initiated a new era of “big data” research in various complex diseases of still-undetermined cause and mechanisms. The investigation of these diseases was, until recently, limited to traditional in vitro and in vivo biological experimentation, but a clear switch to in silico methodologies is now under way. This review tries to provide a comprehensive assessment of state-of-the-art knowledge on omes, omics and multi-omics in inflammatory bowel disease (IBD). The notion and importance of omes, omics and multi-omics in both health and complex diseases like IBD is introduced, followed by a discussion of the various omics believed to be relevant to IBD pathogenesis, and how multi-omics “big data” can generate new insights translatable into useful clinical tools in IBD such as biomarker identification, prediction of remission and relapse, response to therapy, and precision medicine. The pitfalls and limitations of current IBD multi-omics studies are critically analyzed, revealing that, regardless of the types of omes being analyzed, the majority of current reports are still based on simple associations of descriptive retrospective data from cross-sectional patient cohorts rather than more powerful longitudinally collected prospective datasets. Given this limitation, some suggestions are provided on how IBD multi-omics data may be optimized for greater clinical and therapeutic benefit. The review concludes by forecasting the upcoming incorporation of multi-omics analyses in the routine management of IBD.

## 1. Introduction

The investigation of “omics” is now pervasive in the biological and medical arenas, and omics are no longer “fishing expeditions” as they were labeled not too long ago [1]. As it happens in any new field of science, there is great enthusiasm about the potential of omics studies, all the anxiously awaited breakthroughs they might bring and their ensuing applications, which will allow us to better grasp disease pathogenesis and develop brand new therapies. However, expectations from new methodologies are often excessive [2], and they must invariably be toned down by realities related to unexpected difficulties, limitations or pitfalls in their execution, the recognition of actual usefulness, the cost of application of new methodologies, the lack of mathematical, statistical and analytical know-how, and the uncertain interpretation of the results obtained [3,4,5]. This is particularly true when omics are applied to complex biological conditions like chronic inflammatory or neoplastic diseases and the envisioned development of better diagnostic and therapeutic tactics [6].

This review will use inflammatory bowel disease (IBD) as a prototypical example of a complex clinical condition to illustrate and discuss multiple aspects of omics studies carried out in ulcerative colitis (UC) and Crohn’s disease (CD), preceded by a short introduction to omes, omics and multi-omics. The review will discuss multi-omics in health, complex diseases and the potential contributions of multi-omics-derived new data to advance IBD. More importantly, the review will selectively but critically analyze some of the most recent literature on IBD multi-omics, with particular emphasis on the shortcomings of current omics and multi-omics studies resulting from the evaluation of single omes, the arbitrary selection of the omes and the grouping of multiple omes without functional integration. The review will discuss inappropriate or incomplete analysis design, as well as the too-frequently incorrect or cursory conclusions based on superficial interpretations of omics results, and the alluring but often improper use of words like “biomarker”, “prediction”, “association” and “correlation”. Finally, the review will come to an end by concluding that the theoretical potential of IBD multi-omics investigation and the breakthroughs expected to decisively advance understanding of disease mechanisms while opening brand-new therapeutic avenues in UC and CD are not close to being realized.

## 2. Omes, Omics and Multi-Omics

Essentially all diseases, even those not formally pertaining to the category of “complex diseases”, are multifaceted because of the huge amount of varied interconnected biological components and events involved in their pathophysiology [7]. With the appreciation of the innumerable components of any disease came the realization that the study of any single component in isolation would never solve the intricacies of the underlying biology. From the above realization emerged the need to study all components in a far more comprehensive way, hence the emphasis on “omes” and “omics”, defined, respectively as the totality of any particular field and their study. Although the analysis of single omics can provide relevant information, the integration of several omics, i.e., “multi-omics”, allows a much more comprehensive view of the mechanisms underlying any disease [8,9]. All of the omes believe to be involved in IBD are highly complex and variable [10,11], and the investigation of any ome alone can barely open a narrow window into the intricate mechanisms underlying UC or CD. This is as well exemplified by the numerous but insular genomics studies of IBD, where early and subsequent massive DNA sequencing analysis from several thousand patients identified hundreds of common and rare genetic variants associated with CD or UC without generating clear-cut or definitive clues to their etiology, pathogenesis, classification, diagnosis or therapy [12,13,14]. In fact, no clinically useful IBD genetic markers have been identified so far, not even for the variants of *NOD2* that are closely associated with ileal CD at the clinical level without influencing the response to therapy [15,16,17]. After a plethora of reports on the genetics of IBD, there is still no evidence that assessing genetic factors independently of other contributing factors can provide a practical guide for diagnosis or therapy of CD or UC [17]. Given this evidence, and the vast amount of literature available to the reader, the topic of genomics will not be specifically addressed in this review. However, to gain a truly comprehensive understanding of IBD, it is still mandatory to include the genome in combination with other omes, hence the need for “multi-omics” studies carried out with integrative computational methodologies [18].

## 3. Multi-Omics in Health

While the complexity of any disease status is an accepted reality, it is often forgotten that health is also an extremely complex condition [19]. The maintenance of health, with all the endless challenges imposed by growth, environmental factors, adaptation, nutrition, metabolism, immunity, behavior, and mental and physical demands, requires a perfect coordination of innumerous physiological functions and their respective omes [9]. However, the study of health multi-omics lags far behind that of disease multi-omics, hindering and delaying a better understanding of pathophysiological events resulting from the derangements of healthy omes [20,21,22].

The advent of multi-omics is connected to and strictly dependent on new technologies that allow a detailed molecular analysis of omes [23]. The best example is the use of next-generation sequencing (NGS) of DNA with the purpose of defining the complete human genome with its many variants [24]. Sequencing platforms are now used to dissect the epigenome, transcriptome and metagenome, while mass spectroscopy platforms are used to study the proteome and the metabolome [25]. However, it is essential that all omes relevant to both health and disease are studied in a coordinated and integrated way, a fundamental point that will be discussed in greater detail later on in this review.

## 4. Multi-Omics in Complex Diseases

The field of oncology has pioneered most of the omics technologies used for the investigation of complex diseases. Starting in 2006, The Cancer Genome Atlas (TCGA) research network, with the participation of investigators worldwide, has profiled 33 different types of human tumors and discovered numerous molecular abnormalities at the DNA, RNA, protein and epigenetic level associated with distinct cancer types [26]. This landmark approach launched what has now become the conventional approach to studying multiple non-cancerous complex diseases in various human organs and systems. Regrettably, progress with these autoimmune or chronic inflammatory diseases has been much slower than with cancer due to lack of universal coordination and collaboration and more limited resources. Nonetheless, numerous reports of omics or multi-omics studies have been published for a variety of diseases, like rheumatoid arthritis, systemic lupus erythematosus, asthma, chronic obstructive pulmonary disease, and even irritable bowel syndrome and autism [27,28,29,30,31,32]. It should be kept in mind that these complex diseases occur in vastly different organs and tissues that have their own tissue-specific regulatory circuits in both health and disease, and these differences will be reflected in the respective multi-omics analyses [33].

## 5. Multi-Omics in IBD

The importance of omics in intestinal inflammation and their potential for advancing disease understanding, biomarker discovery and new therapies have been recognized since the mid-2000s [34], but the appreciation of omics by the medical community has been sluggish and only recently have studies begun to be undertaken and generate some tangible results. Though not to the same degree of the TCGA network, efforts to create working groups for the global study of omics and multi-omics in IBD have also been made, like the 1000IBD project (https://1000ibd.org, accessed on 10 August 2023) [35]. The project recruits and prospectively follows adult patients with new or existing IBD of different phenotypes and collects information about several omes (exposome, genome, transcriptome and metagenome, drug response, etc.). The 1000IBD project has generated a number of publications on various topics, like the modulation of intestinal gene expression by inflammation [36], the association of dietary patterns with selective features of the gut microbiota [37], the impact of drugs on gut microbiota composition and function [38], the identification of environmental factors linked to IBD development [39], the association of IBD genotypes and phenotypes with the plasma proteome [40], as well as reviews on multi-omic data availability in IBD [41], and several other aspects of IBD pathophysiology. Another similar but more recent group initiative is the European ImmUniverse Consortium (https://www.imi.europa.eu/projects-results/project-factsheets/immuniverse, accessed on 8 January 2019) which was formed to create a multi-omics integrative approach to personalized medicine in immune-mediated inflammatory diseases, including IBD [42]. Several other IBD omics-oriented initiatives are also under way worldwide [43].

However, the vast majority of reports on IBD multi-omics are still the result of work carried out by individual investigators and, whether as reviews or original reports, have substantially expanded UC and CD omics information on a wide variety of topics. Reviews tend to emphasize the importance of multi-omics in IBD for the goal of delivering more effective treatment [44] or precision medicine to UC and CD patients [45,46], or the potential of multi-omics to identify predictive biomarkers for the discovery of IBD risk factors [47], while other reviews call attention to the numerous challenges of designing and analyzing multi-omics IBD data [48], the latter being a key topic to be further and comprehensively elaborated upon later on in this review.

The number of omics reported in the literature at large or listed in formal databases is large (Table 1). However, what omics are relevant to IBD, to what degree, which ones should be prioritized for study, which ones should be functionally integrated, and which omes research should be our main focus is a critical and unresolved issue.

Until recently, four major omes received the bulk of the attention from IBD investigators: the exposome, the genome, the microbiome and the immunome (Figure 1, left panel). With the recent advent of the sequencing and mass spectroscopy platform, the type and number of omes accessible to IBD-related investigation have drastically increased, including the epigenome, transcriptome, proteome, metabolome, etc., but many additional unexplored omes are undoubtedly relevant to CD and UC pathophysiology (Figure 1, right panel).

### 5.1. Exposomics

The study of the environmental factors (the exposome) as they related to IBD overall, or IBD pathogenesis in particular, is the most difficult and problematic because of the endless number of factors involved, their multiplicity of actions, the constant generation of new man-made substances that permeate the environment at large and modify the exposome, the individuality of the exposure process by humans and the unique response of each person [49,50,51]. Reports and reviews on the assumed or demonstrable impact of environmental risk factors in IBD are plentiful [39,52,53], but it is logistically and practically impossible to identify all relevant factors in each IBD patient and derive usable omics data to be factored in for multi-omics integration.

### 5.2. Microbiomics

The importance of the gut microbiome to IBD pathogenesis has been long established [54] and progressively explored in greater depth with the adoption of metagenomic analyses [55]. Studies analyzing the composition and function of the gut microbiota in UC or CD patients are numerous, but when performed in isolation they have generated unclear and inconsistent results [56]. However, metagenomic analyses of the gut microbiome combined with other omes can result in more relevant information about the role of microbes in the pathobiology of IBD. For instance, a study performed as part of the Human Functional Genomics Project explored the combination and functional relationship of multi-omic microbial and cytokine profiles and found a link between the human gut microbiome and inflammatory cytokine production [57]. Using longitudinally collected blood, stools and biopsy samples, an IBD multi-omics (metagenome, metatranscriptome, proteome, metabolome, and virome) database has been created, revealing that patient disease activity is marked by temporal increases and shifts in taxonomy, as well as functional and biochemical composition [58]. The combination of metagenomics, metatranscriptomics and metaproteomics can uncover multiple microbial bioactive molecules likely to interact with the host immune system in IBD, and inform us about the broad and dynamic host–microbial interactions involved in IBD pathogenesis [59]. While enlightening and exciting, all results of microbiomics reports must be interpreted with caution in view of the extensive impact of genetics [60], lifestyle [61], nutrition [62], immunity [63], common drugs or drugs used to treat IBD patients [38,64] on gut microbiota composition and function.

### 5.3. Immunomics

Immunology has dominated the study of IBD for a long time, and only in the last decade or so has its dominance partially diminished due to a growing interest in the gut microbiota and other non-immune factors in IBD pathogenesis [65]. The key role of the immune response in IBD is unquestionable, but this response is influenced by a myriad of factors such as the host, environment, genes, microbes, nutrition, behavior, season of the year, stress, etc. [51,57,63,66,67]. This multiplicity of unpredictable and uncontrollable factors puts constant demands on the immune system, which responds by continuously adapting in a vigorous way, leading to both beneficial and harmful effects [68]. This multidimensional response makes it unrealistic to expect that the immune system alone can provide reproducible and reliable markers or omics that define specific biological states in multifactorial diseases like CD or UC. A good example is a report which claimed that the transcriptional signature of circulating CD8+ T cells could predict the disease course of patients with CD and UC [69], a claim that could not be reproduced in a subsequent study [70]. These contrasting results are likely be due to the plasticity of CD8+ T cells in IBD [71], a quality inherent to all immune cells residing in specific tissue microenvironments [72], which makes them and their products, like cytokines, essentially impossible to use as predictable and reproducible qualitative or quantitative measurements of any immune response [73]. Fundamental abnormalities of the immune response are clearly involved in the pathogenesis of UC and CD and, considering the variability and unpredictability of the immune response, such abnormalities cannot be evaluated in isolation, but integrated with multiple other omes under the umbrella of the modern comprehensive notion of the IBD interactome [74].

### 5.4. Epigenomics

The field of epigenetics, variably defined as DNA-independent changes in gene expression or the transgenerational effects and/or inherited expression states [75], is a more recently and relatively less investigated omic in IBD [76], but it is of crucial importance because of the link it establishes between genes and the exposome [77]. Intestinal biopsies obtained from pediatric CD and UC patients submitted to combined DNA methylation and transcription analyses were able to identify disease subtypes and an association with clinical outcome [78]. A study by Kalla et al. performed multi-omic integration of the methylome, genome and transcriptome measured in the peripheral blood of UC and CD patients and identified multiple differentially methylated positions in IBD with an association with treatment escalation or need for surgery [79]. The latter group of investigators recently reported a differential and variable methylation in CD patients developing clinical recurrence after surgery [80]. The study of epigenetics in IBD is now attracting more and more attention, but it also must be considered in the context of interactions with the genome and the exposome [81].

### 5.5. Proteomics, Metabolomics, Lipidomics

The possibility of using serum proteomics for biomarker discovery in IBD has been considered for a long time [82], and this approach, particularly when combined with metabolomics and lipidomics, could help us to discover complementary biomarkers, monitor and predict response to therapy and promote personalized medicine [83,84]. The combined alterations in serum lipids, amino acids and energy metabolites have been reported to distinguish UC from CD and healthy controls [85], while a study by Fan et al. compared plasma lipid profiles in UC vs. CD patients and found that a number of ether lipids were negatively associated with CD [86]. Another study reported that neither metagenomic or host genetics could distinguish ileal and colonic CD, but this could be accomplished via metabolomics and metaproteomics analyses using mass spectroscopy [87].

### 5.6. Single-Cell Technologies, Omics, Multi-Omics and Spatial Multi-Omics

With the progress prompted by the development of many new cellular and molecular tools complemented by the even more rapid deployment of artificial intelligence, machine and deep learning analytical methods, the possibility of studying omics and multiomics at the single-cell level has become an exciting reality [88]. An enormous amount of literature has rapidly accumulated on single cell omics, multi-omics and spatial transcriptomics in different cells and organs. Initial studies focused primarily on single-cell transcriptomics [89,90], resulting in the universal and fundamental discovery of an unsuspected extreme degree of cell heterogeneity of both immune [91,92] and non-immune cells [93,94] in essentially all tissues, including the gastrointestinal tract [95,96,97]. These reports were then followed by single-cell multi-omics studies in which the genome, epigenome, transcriptome, proteome, or other omes could be assessed in the same cells [88] and the subsequent investigation of single-cell multi-omics, i.e., the detection and topographical mapping of multiple omes in specific tissue types [98] in both normal and inflamed tissues [99]. The field of single cell analysis, omics and multi-omics is still evolving very rapidly and in the near future, reference maps of the whole human body will become available with an unprecedented level of cellular and molecular precision [100].

In the field of gastroenterology and IBD in particular, similar studies have emerged that also show vast single cell heterogeneity in immune and epithelial cells in both UC and CD [101,102,103]. Recently, additional studies have appeared describing fibroblast heterogeneity and cellular interactions of a full thickness single-cell transcriptomic atlas of the strictured CD intestine [104], and the single-cell spatial proteomics and transcriptomics analysis of colonic biopsies of patients with UC receiving treatment with vedolizumab [105].

## 6. Clinical Applications of Multi-Omics Analyses

The primary goal behind the study of multi-omics in IBD is the discovery of novel tools that afford more precise diagnostic and management means at the bedside. Based on this very desirable goal, a large and growing number of studies have been and are still being published which claim that the utilization of multi-omics and their different combinations can help with the discovery of basic, translational, and clinical elements of UC and CD. The execution of these studies is justifiable and the results are of interest. However, most of the claims so far made must be carefully assessed and interpreted in the light of more stringent criteria, as discussed later under the subtitle Pitfalls and Limitations of Current IBD Multi-omics Studies.

### 6.1. Biomarker Identification

The quest to identify disease biomarkers is as old as the field of medicine. The popular biomarkers commonly used for the evaluation of UC and CD patents, like C reactive protein and fecal calprotectin, are not true disease biomarkers as they simply reflect the process of inflammation and its degree, but are not specific and their levels increase as a result of many other conditions [106]. The same can be said for serological biomarkers, whose combined evaluation is of uncertain value for the prediction of disease progression or response to treatment [107]. In 2016, the National Institutes of Health (NIH) and the Food and Drug Administration (FDA) created the FDA-NIH Biomarker Working Group with the goal of better defining what is intended for “biomarker” and their many types; this was last updated in 2021 [108]. The FDA-NIH working group identified the following types of biomarkers: susceptibility/risk, predictive, diagnostic, monitoring, pharmacodynamic/response, safety, prognostic as well as “reasonably likely surrogate endpoints” [108]. In principle, all of them are of interest in IBD and the topic has generated a great deal of attention and a huge volume of publications [109]. The possibility of biomarker discovery using omics and multi-omics approaches has been a goal for multiple inflammatory and neoplastic conditions [110,111,112,113,114], including IBD [115], but so far precise and definitive biomarkers have yet to be found.

### 6.2. Prediction of Remission and Relapse

The identification of biomarkers that predict clinical relapse or clinical remission of IBD patients using multi-omics profiling has been the target of numerous studies. One study based on profiling of the blood proteome, metabolome and microbiome claimed that a proinflammatory state predisposing to clinical relapse could be identified in UC and CD patients [116], while another study carried out tissue and blood plasma proteome and plasma metabolomics profiling of UC patients and concluded that biomarkers indicative of functional remission could be identified [117]. A very recent report found that certain blood protein profiles were associated with the risk of clinical relapse in patients who stopped infliximab therapy [118], a potentially useful solution to the vexing issue of when to stop this common form of biological therapy. None of these studies have been replicated or validated until now and their value remains unclear.

### 6.3. Response to Therapy

In an early pilot study, serum proteomics were used to establish a model to predict response to infliximab treatment, with the results suggesting that response was associated with platelet metabolism [119]. A subsequent study based on prospectively collected blood and stool samples carried out metagenomic, serum metabolomic and serum proteomic measurements and found that some microbial factors affected the response of IBD patients receiving biological therapies [120]. A more recent study by Mishra et al. performed a longitudinal, blood-based multi-omics (RNAseq and genome-wide DNA methylation) study in two prospective IBD cohorts given TNF antagonists to predict therapy response, but no consistent predictive molecular signatures could be found [121]. Thus, the prediction of response to therapy in IBD patients based on results of multi-omics analyses appears to be at a preliminary stage and the results vary depending on the number and combination of the omics included in the prediction modeling. Perhaps more importantly, it still remains to be verified whether any of the newly described omics-based biomarkers are indeed better than traditional ones, like C-reactive protein [122].

### 6.4. Precision Medicine

The implementation of precision medicine is the ultimate goal behind the study of multi-omics in complex diseases [123]. Precision medicine has many definitions, the one officially released by the NIH being one of the most comprehensive and commonly cited: “*An emerging approach for disease treatment and prevention that takes into account individual variability in genes, environment, and lifestyle for each person*” (www.nih.gov/precisionmedicine, accessed on 10 August 2023). The need for and importance of developing precision medicine are undeniable and have an enormous potential, but they require the integration of clinical data with multiple molecular profiles, and this is only possible with the most advanced computational methods and a close collaboration of clinicians, researchers and bioinformaticians [124]. With the advent of NGS and various other technologies that allow molecular characterization of health and disease, the quest for precision medicine in IBD has also been relying on omics and multi-omics studies for its fulfilment. This goal is extremely challenging and has yet to be achieved [125], but various publications reinforce the fundamental importance of pursuing multi-omics studies [126,127], including a recent series of state-of-the-art reports by the European Crohn’s & Colitis Organization (ECCO) [11,128,129,130].

## 7. Pitfalls and Limitations of Current IBD Multi-Omics Studies

The field of medicine is becoming increasingly dependent on technological progress, and it is no surprise that many hopes and expectations for a better understanding and management of IBD now count on omics and multi-omics tactics [131]. However, what it is often forgotten is that these advanced molecular approaches are being applied to diseases of extraordinary biological complexity like CD and UC, which makes the application of the recent artificial intelligence-based tools inconsistent and the interpretability of results challenging [132]. Thus, if we objectively evaluate the current literature on IBD omics, it seems naïve and unrealistic to expect that the multi-omics studies performed so far can provide truly groundbreaking advances in IBD, at least for the time being. This still unsatisfactory contribution to clinically useful advances in IBD is partly due to a number of factors that are seldom taken into account, such as biological complexity, human variability, study design, technical oversights, and overinterpretation of results, some of which are discussed below.

### 7.1. Human Complexity and Variability

Healthy humans represent extraordinarily diverse groups of people, each group having its own peculiar historical, topographical, genomic, environmental, proteomic, metabolomic, nutritional, microbial, behavioral and evolutionary characteristics [133,134]. This diversity persists when humans become ill and impacts all their biological functions like, for instance, the production and abundance of protein levels, which are heritable molecular phenotypes that exhibit considerable longitudinal variation between populations, individuals and sexes [135,136]. If we consider this protein diversity in an inflammatory response, like in IBD, it is easy to envision how levels of many inflammatory products, like cytokines for instance, will also vary considerably from patient to patient and from time to time during disease evolution, making very difficult (if not impossible) to establish what is a “high”, “normal” or “low” value of any cytokine or any other inflammation-driven product. A consequence of this overwhelming variability leads to different endophenotypes of complex diseases [137], and when their molecular counterparts are used for omics and multi-omics evaluation the variability will be embedded in the analysis and unsuspectingly modify the results and the ensuing interpretation. Complexity and variability are key features of IBD that must be recognized and taken into account in multi-omics interpretation, otherwise the conclusions reached will be incorrect or misleading [10].

### 7.2. Exclusion of Influential Modifying Factors

In addition to the inevitable biological human diversity and its impact on multi-omics studies, a series of other factors affect and modify the results, reproducibility and interpretation of multi-omics studies of IBD. These factors are usually not cited, not considered or discounted by the investigators, becoming “unrecognized variables” that may substantially affect the results [138].

One example of such factors is the environment to which the patient was exposed at birth, and thereafter during his or her subsequent lifetime [49]. Host genetics have a relatively minor influence on the microbiota, while diet, drugs, and the environment at large have a much greater capacity to shape the gut microbiota which, in turn, is a key determinant of the immune response, both in health and inflammation [36,53]. Prior environmental exposures of UC and CD patients and the environment at the time of their inclusion in an omics and multi-omics investigation are hardly ever recorded, mentioned, or taken into account, resulting in the exclusion of information on a component essential for the correct outcome and interpretation of the study.

As previously mentioned, an abnormal immune response is an intrinsic component of CD and UC, but other factors, extrinsic to IBD but intrinsic to the host (the IBD patient), that significantly modulate the immune response, remain unknown and therefore are not included in multi-omics analyses. A good example is chemical pollutants. Chemical pollutants are ubiquitously present, and many of them exert negative effects in both humas and animals, like, for instance, perfluorinated compounds [139]. These compounds alter inflammatory responses, the production of cytokines and adaptive and innate immune responses by altering the glycosylation status of the immune cells, which regulates their function both in health and disease [140]. Glycans are carbohydrate sequences that are added to proteins or lipids and modulate their structure and function [141], as is the case for T-cell responses, which have been shown to be considerably altered by glycans in IBD [142]. Despite its importance, the glycosylation status of the immune system of IBD patients is hardly ever considered, and the impact of this vital variable is not reflected in the results of the omics analysis.

Other examples of modifying factors that are almost never assessed, or are ignored, and therefore never taken into account in multiomics analysis include the patients’ dietary patterns [37], their stress status [66], and even their sex, considering that significant sex differences in the transcriptome and its genetic regulation occur in humans [143].

All the modifying factors mentioned in this section most likely represent only a handful of the countless other modifiers that exist but are never included in multi-omics analyses. Most of these modifying factors are not identifiable in the experimental or validation cohorts from which data are obtained prior to the final analysis and cannot be factored in the results; nevertheless, they most likely skew the final outcome and its interpretation. Nevertheless, this stark reality should not discourage us from the use of multi-omics studies but should be always kept in mind to instigate caution and reservation when interpreting multi-omics studies.

### 7.3. Cross-Sectional Data, Different Omics Combinations and Biological Commonalities

Several other pitfalls currently limit the accuracy of essentially all IBD multi-omics studies, pitfalls that are obvious but almost never mentioned or openly discussed in the literature.

A particularly important one is the almost exclusive cross-sectional nature of the study design. The sophisticated analytical tools used for multi-omics analysis and integration usually hinge on biosamples obtained from individual subjects at a single time point, in general as well as in IBD reports. Regardless of whether blood, serum, plasma or tissue are used as biosamples, of whether a single or several omes are included in the study and of the sequencing or mass spectroscopy technology employed [25], the biosamples will only reflect the biology of that particular time point but not the complete biology of chronic continuously evolving conditions like CD or UC. Chronic diseases like IBD share many common and non-specific biological pathways with innumerous other diseases, and these pathways can be readily detected using modern molecular and computational tools. How well these shared pathways reflect the biology relevant to each specific condition can be difficult to discern. An intriguing example of this difficulty is illustrated in a recent publication reporting a shared immune pathobiology between IBD, asthma and COVID-19, and similarities of disease manifestation associated with COVID-19 in IBD, stroke and attention-deficit hyperactivity disorder by network proximity measure [144]. IBD, COVID-19, asthma, stroke and attention-deficit hyperactivity disorder hardly share any clinicopathological manifestations, but nevertheless they appear to share common biological elements. The above conundrum brings up the several critically important questions listed in Table 2:

With the exception of the first two questions (different tools, different results, biological specificity), the likely answer to all other questions is almost certainly a “no” because of the mentioned cross-sectional one time-point only design of the study, in addition to the previously cited human variability [133]. The “no” brings up another critical point: can we trust these selectively collected results to give us a correct representation of the pathobiology of IBD or should we be carrying out these studies in another more trustworthy way?

### 7.4. Single-Cell Omics and Multi-Omics: Cell Isolation Pitfalls

The advent of molecular technologies that allow the investigation of omics and multi-omics at the single-cell level brought in new analytical perspectives of IBD, but also some drawbacks intrinsically associated with any cell isolation process. Tissue dissociation and cell isolation processes have been used in IBD research for a long time [145], and have been instrumental to a better understanding of cellular and molecular mechanisms of disease, and recent studies describing the organization of the human gut at a single-cell resolution will certainly help to expand knowledge of the normal and inflamed intestine [146]. With the adoption of tissue and cell manipulation processes for single-cell omics and multi-omics, old and new methodological pitfalls have materialized, which unfortunately are largely dismissed in most publications, even though they can clearly affect the results of omics and multi-omics studies.

There are multiple ways to isolate cells from liquid and solid tissues in IBD, primarily the peripheral blood and intestinal tissue derived from mucosal biopsies or surgical resection, and all of them introduce minor or major artifacts that are practically impossible to eliminate. Even in the absence of harsh enzymatic or mechanical steps, like those required during blood cell isolation, loss of cell surface receptor expression can occur, and this can affect receptor-dependent functions [147]. Loss of epitopes is even more frequent when enzymatic digestion is used [148], as it is routinely the case with intestinal tissue. Especially relevant to omics studies is the observation that loss of cellular function, cell–cell interaction and disruption of tissue architecture can affect transcriptional events [149], and this would obviously introduce artifacts that alter omics measurements.

Cell isolation from solid tissue, like from any segment of the gastrointestinal tract, normal or IBD-involved, never fully translates the complete in vivo cellular composition, and there is always some degree of non-specific as well as selective cell loss of both quantitative and qualitative nature. Therefore, when the resulting single-cell preparations are utilized for omics or multi-omics studies, the results will not be fully representative of all the in vivo molecular events. Cell heterogeneity will be detected, but it will not be exactly the same as that in the original tissue sample prior to manipulation [150]. In addition, the intestinal microenvironment is highly dynamic, particularly in the immune compartment, and we do not know whether the cellular, molecular and omes ex vivo composition obtained at any given time point is the same at other time points before or after the single-cell isolation process. A related crucial issue is the interpretation of what and how much single-cell findings represent: a highly cited publication by Martin et al. describes the detection of a pathogenic cellular module (GIMATS) in CD lesions associated with resistance to anti-TNF therapy [101]. However, GIMATS was only identified in a subset of the CD patients, which begs the question of how many other modules may also exist in the same or other patient cohorts, and which modules are directly or indirectly relevant to CD pathobiology and therapeutic response. Each module will obviously have its own omics and multi-omics make up characterized by other single-cell subsets, but their relevance to CD pathobiology remains unclear.

As molecular, omics, analytical and barcoding methodologies continue to advance and become increasingly common, the number of reports on single-cell omics and multiomics will inflate and generate more and more data on single-cell omics and multi-omics [98]. It is unlikely that the cell isolation pitfalls discussed in this section will ever be addressed in a systematic way and attempts will be made to routinely minimize them, and thus what remains is the hope that investigators will be vigilant and aware of potential or probable misleading omic data.

## 8. Optimizing the Use IBD Multi-Omics Data

The widespread availability of private and commercial computational platforms that can evaluate millions of data points from genomics, epigenomics, transcriptomics, proteomics, metabolomics, metagenomics and other omics sources has greatly facilitated access to disease “big data” by researchers and clinicians alike [3], including in IBD [43]. This has generated massive amounts of reports which make all sort of claims about what the omics or multi-omics data show and their practical or theoretical biological, translational or clinical worth, particularly with regard to biomarkers, outcomes and response to therapy, as mentioned previously in this review. Unfortunately, many of these claims are not firmly substantiated by high-quality data and analyses, and results are often overinterpreted beyond what the data really show. In the following sections, we will try to offer suggestions on how to correctly interpret the omics and multi-omics data in light of scientifically accepted definitions and criteria.

### 8.1. Biomarker Validation, Reproducibility and Predictive Value

The identification of clinically reliable biomarkers and the prediction of remission, relapse and response to therapy rank among the top expectations that the clinical IBD community eagerly awaits from multi-omics studies [43]. Biomarkers that could clearly separate CD from UC and their multiple subtypes would be hugely valuable in daily practice, particularly if associated with biomarkers that could help make therapeutic decisions with a high probability of success. The recent IBD literature is full of reports claiming that such biomarkers have been identified, but most reports do not include clear definitions of biomarker, nor do they use strict criteria. To be reliable, any biomarker must meet strict standards for validation and reproducibility [138].

Validation means that the proposed biomarker is also detectable in separate control cohorts composed of the same patient phenotypes with similar clinical, therapeutic and evolutionary characteristics; another source of validation is the replication of the same biomarker in the original patient cohort but at a different time point during disease evolution; the presence or absence of the claimed biomarker in patients without IBD but with a condition of autoimmune or chronic inflammatory nature, like rheumatoid arthritis, psoriasis, etc., would also be very useful to help us learn whether the biomarker is disease-specific or simply translates general pathogenic events. If a biomarker is proposed as a predictor of remission, relapse or therapeutic response, these very same outcomes must be reproduced in the original or validation cohorts at different time points during disease evolution. These are admittedly high bars to meet, but a biomarker must prove reproducible for us to gain the confidence necessary to rely on it in clinical practice [138]. Unfortunately, none of the current reports in the IBD literature meet these validation criteria, so that presently most claims of new biomarker discovery for UC, CD or IBD must be interpreted with caution and only seen as observations to be further explored.

Furthermore, essentially all studies claiming the detection of novel biomarkers in IBD are based on retrospective or only one-time data where an association is found between a particular set of observations and a clinical or therapeutic outcome [151]. The word association is commonly misused to mean dependence or correlation, but association and correlation are distinct parameters, as pointed out by Altman and Krzywinski [152]. Correlation is a specific term that indicates that two variables display the same linear upward or downward trend, which is never the case for highly variable clinical variables, and is measured with the Pearson’s correlation (*r*) test; on the other hand, an association, the term most commonly found in IBD publications in which the authors claimed to have discovered a new biomarker, is only a hypothesis without proof and does not imply causation [152]. Importantly, when the multiple variables are evaluated in complex systems, like IBD, spurious associations are common, casting additional doubts about the validity of new biomarker discovery claims.

### 8.2. The Power of Longitudinal Multi-Omics

IBD is a prototypical example of a chronic, often life-long condition which displays disparate and variable clinical manifestations during its evolution; these are brought about by unpredictable changes in the causal pathogenic mechanisms. Although the heterogeneity, variability and unpredictability of CD and UC are well known, there are lamentably few longitudinal studies of IBD patients that systematically evaluate multi-omics changes over time, creating a knowledge vacuum that hinders and delays the improvement of the presently unsatisfactory forms of IBD management. As underscored on multiple occasions in this review, healthy and diseased humans are highly heterogeneous, a fundamental characteristic maintained throughout a person’s lifetime [133,135,136]. Thus, profiling multi-omics over time in a defined IBD patient population would be extremely valuable, providing a temporal flow of information about the cellular and molecular changes preceding flare-up and remission events, and perhaps even predicting when and how to intervene to avoid disease recurrence.

Few but very informative longitudinal studies of other diseases have appeared in the literature. An early study of an individual with type 2 diabetes followed and multi-omics-profiled during periods of health and disease revealed extensive, complex and dynamic changes in omics profiles [20]. This approach is feasible for healthy as well as sick people, and in IBD it could explain individual differences between the patients and how to optimize therapies [22]. The performance of an integrated multi-omics profiling is logistically difficult, laborious and costly, but it could open the door to an unprecedented deep level of disease understanding, improved healthcare and the actual implementation of precision medicine [153].

Given the paucity of human IBD longitudinal studies with prospective monitoring of multi-omics, this type of follow up can be performed in animal models of IBD, and some reports provide clear evidence of major cellular and molecular shifts during progression of intestinal inflammation. Interleukin (IL)-10-deficient mice display a Th1 colitis with elevated production of IL-12 and interferon (IFN)-γ in the early phase of disease, while in the late phase there is a progressive increase in the Th2 cytokines IL-4 and IL-13 [154]; similarly, in the SAMP1/YitFc model of ileitis, also a Th1 model of intestinal inflammation with an early elevation of IFN-γ and TNF-α production, the Th2 cytokines IL-5 and IL-13 significantly increase in the chronic stages of inflammation [155]. The relevance of these cytokine shifts to mediation of inflammation was demonstrated via the amelioration of colitis in IL-10-deficient mice by IL-12 neutralization in the early but not late phase of colitis [154], while in SAMP/YitFc mice, blockade of CCL25/CCR9 effectively improved inflammation in early but not late ileitis [156]. So, at least in experimental IBD, there is strong evidence that fundamental changes in functional immunomics occur with disease evolution.

### 8.3. High Level Integration

Given the complexity, variability and heterogeneity of IBD, the need to think in terms of an IBD interactome and create integrated multi-omics molecular profiles in IBD is obvious and has been stressed in multiple publications [10,44,74]. This is not only true for IBD but any other condition where personalized molecular profiles are being sought and, as pointed out by McCarthy and Birney, these profiles must capture all combined facets of health or disease [157], hence the need for multi-omics integration.

There are large international biomedical databases from which to obtain vast omics information, like, for instance, the British UK Biobank (https://www.ukbiobank.ac.uk, accessed on 10 August 2023) or the Chinese National Genomics Data Center [158]. Alternatively, investigators may access the biobanks in their own countries [43] or develop and customize their own databases. However, each database has its own structure, biological features and data type, and these must be taken into account when retrieving omics data for subsequent integration. On the other hand, while each omics database has its own structure, they essentially all use the same data standard. FASTQ files, text-based formats for storing a biological sequence data and its quality score, follow the same standards in the Chinese National Genomics Data Center as well as in the NIH Sequence Read Archive (SRA) database, or the European Nucleotide Archive (ENA) collection. The same is true for other formats of gene expression like the Binary Alignment Map (BAM), Variant Call Format (VCF), Transcripts per Kilobase Million (TPM), and the genome is generally the same (the current standard is the human genome version hg38 (https://www.ncbi.nlm.nih.gov/datasets/genome/GCF_000001405.26/, accessed on 26 September 2023)

There are literally hundreds of computational tools that can be used to analyze and integrate multi-omics data [4,5,9,18,159,160], but the use of any of these tools still poses several obstacles. To start, patient-derived omics data are never normally distributed along Gaussian curves, only randomly distributed, and there are no perfect ways to analyze these data [25]. Another obstacle is the choice of the most appropriate computational approach to be used [161], and the specific goal of the study and question(s) being asked should determine the selection of the tools to be employed for multi-omics integration. As pointed out by Amagah et al., these tools are usually different if the goal is to identify molecular drivers of disease, determine biological processes or disease phenotype, predict outcomes or disease progression or to facilitate drug discovery [18]. For instance, Sudhakar et al. used multi-omic factor analysis (MOFA), iCluster, Group factor analysis (GFA), CEMiTool, CheA, Parsimonious Composite Network (PCNet) and Reactome as the integration tools for understanding the molecular drivers of disease heterogeneity in CD [162], but these same tools may not be the best for network identification, disease phenotype or drug discovery.

Sample size is another important factor and, in general, the larger the better [163], but a few hundred biosamples, rather than thousands, are sufficient to generate interpretable and reliable results, as demonstrated by the TCGA Consortium [26]. In general, it is better to have a higher number of omics datasets with an average number of biosamples rather than a very large number of biosamples but few omics datasets, as this improves accuracy and facilitates interpretation [30].

In addition to the size of the biosamples and number of omics, how data are collected is also critical. Data from cross-sectional studies, by far the most commonly reported in the recent IBD omics literature, do not allow randomized interventions and preclude causal inference [25]. On the other hand, as already highlighted in the section on the power of longitudinal multi-omics, prospectively and repetitively collected omics data are far superior to cross-sectional data [21,153,164], but are also more difficult to obtain.

Finally, another considerable obstacle is the reality that the IBD community, which is the one to most benefit from detailed and correct multi-omics analyses, is largely unfamiliar with omics and the machine or deep learning methodologies necessary for their analyses. This requires collaboration with computational biologists, who may or may not be available or readily accessible to investigators or practitioners, including the creation of multidisciplinary research teams (physicians and basic scientists) to take full advantage of the power of multi-omics in IBD [165].

## 9. Conclusions and Future Directions

The discussions and considerations presented in this review lead to some fundamental conclusions and a few realistic expectations. The first conclusion is obvious, i.e., multi-omics studies in IBD represent an exciting and rapidly expanding field of investigation that is badly needed to advance disease understanding and go beyond the still popular but erroneous notion that a single IBD pathogenic component can be dominant, as it has been for immunology and genetics until recently and now for the microbiome.

Another fact-based conclusion is the clear evidence that, so far, the vast majority of omics and multi-omics reports in UC or CD are simple cross-sectional studies that are basically descriptive in nature and largely carried out in single patient cohorts not corroborated by validation cohorts and functional studies. This approach considerably limits their value and the reliability of the results. Cross-sectional studies will not and should not be abandoned but must be combined with longitudinal prospective multi-omics datasets.

An additional realistic conclusion is that the expectation that multi-omics will yield accurate diagnostic, prognostic and predictive biomarkers to help clinicians in their daily practice is not going to be fulfilled any time soon. Only carefully performed and reproducible multi-omics data in well defined, rather than random, IBD patient cohorts will allow reliable stratification of patients based on molecular subgrouping. This is essential to define clinically useful UC or CD patient subgroups with relatively uniform pathogenic mechanisms that can be individually targeted to promote personalized medicine, and only this will allow us to abandon the current inefficient “one size fits all” therapeutic approach.

A final reality is that the IBD community at large must accept the fact that the existing diagnostic and classification tools will inevitably change to molecular ones, and that computational analyses will progressively and inexorably replace clinical tools such as personal knowledge and experience, endoscopy, histology, imaging and lab tests. As a consequence, therapeutic decisions will be based on molecularly defined targets resulting from multi-omics screening and network analysis rather than the current targets like non-specific single inflammatory products. This will require an educational process to be conducted by the IBD community, and regular collaboration with computational biologists must be viewed not as an imposition, but as a necessity that will make for better doctors.

## Figures and Tables

**Figure 1 ijms-24-14912-f001:**
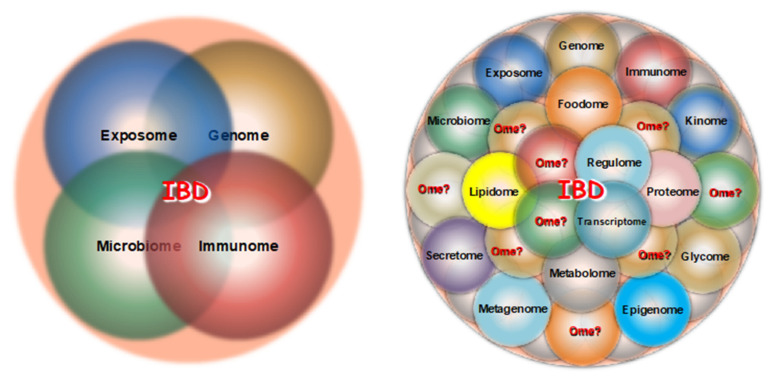
Traditional (**left panel**) and more recent (**right panel**) omes under investigation in IBD.

**Table 1 ijms-24-14912-t001:** Omes and omics glossary.

Type of Ome	*Field of Study* *(Omics)*	Type of Ome	*Field of Study* *(Omics)*	Type of Ome	*Field of Study* *(Omics)*
**Allergernome**	*Allergernomics*	**Immunome**	*Immunomics*	**Pharmacogenetics**	*Pharmacogenetics*
**Bibliome**	*Bibliomics*	**Interferome**	*Interferomics*	**Phenome**	*Phenomics*
**Connectome**	*Connectomics*	**Interactome**	*Interactomics*	**Physiome**	*Physiomics*
**Cytome**	*Cytomics*	**Ionome**	*Ionomics*	**Phytochemome**	*Phytochemomics*
**Diseasome**	*Medicine*	**Kinome**	*Kinomics*	**Proteome**	*Proteomics*
**Editome**	*RNA editing*	**Lipidome**	*Lipidomics*	**Regulome**	*Regulomics*
**Embryome**	*Embryomics*	**Mechanome**	*Mechanomics*	**Researchsome**	*Research areas*
**Envirome**	*Enviromics*	**Metabolome**	*Metabolomics*	**Secretome**	*Secretomics*
**Epigenome**	*Epigenomics*	**Metagenome**	*Metagenomics*	**Speechome**	*Speechomics*
**Exposome**	*Exposomics*	**Metallome**	*Metallomics*	**Toponome**	*Toponomics*
**Foodome**	*Foodomics*	**Microbiome**	*Microbiomics*	**Transcriptome**	*Transcriptomics*
**Genome**	*Genomics*	**Obesidome**	*Obesidomics*	**Trihalome**	*Medicine*
**Glycome**	*Glycomics*	**ORFeome**	*ORFeomics*	**Volatilome**	*Volatilomics*
**Holgenome**	*Holgenomics*	**Organome**	*Organomics*	**Etc.**	*Etc.*

* Adapted from http://www.genomicglossaries.com/content/omes.asp/ (accessed on 10 August 2023). The omes in orange boxes are known or likely to be relevant to IBD.

**Table 2 ijms-24-14912-t002:** Queries, reservations and variables in IBD omics analysis.

(1) Is the detected underlying biology specific to CD or UC or IBD?
(2) Will the results be different if different multi-omics tools are employed?
(3) Do the results obtained with the chosen multi-omics tools represent the overall IBD status of the patient cohort or only the biology of a particular time point?
(4) Will the results be the same if the same biosamples are derived from a separate IBD cohort?
(5) Are the results only representative of a subset of IBD patients?
(6) Are all the biosample-providing patients uniform with regard to IBD phenotype, clinical and histological activity, medications and time of evolution?
(7) Would the multi-omics results be the same if the same types of biosamples from the same patient cohort are obtained at a different time point?
(8) Have age and sex variables been taken into account?

## Data Availability

All data derived from literature cited in PubMed.

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
