# Peer review of "Omics and Multi-Omics in IBD: No Integration, No Breakthroughs"

_ijms, 2023, doi:10.3390/ijms241914912_

Round 1

Reviewer 1 Report

In the review article "Omics and multi-omics in IBD: no integration, no breakthroughs" the use of sophisticated technologies in clinical research of IBD is presented in great detail and critically. the complexity of the research is described in a very interesting way and that we still have a lot of work to do in order to be able to interpret the results and draw correct conclusions. I read the article with great interest and I wouldn't change a thing.

Author Response

Thank you very much for the positive comments.

Reviewer 2 Report

Dear author, 

I have studied with great interest the manuscript ‘’Omics and multi-omics in IBD: no integration, no break-2 throughs’’The text is clear, well organized, and interesting to the reader. However, despite its focus on a novel topic, it is not clear to me what ''new'' information this manuscript provides, given the existence of previous papers on IBD and omics. On the other hand, the manuscript is difficult to read because it has so much text and so few images/tables. It would be advisable to include a graphical abstract. On the other hand, the author could consider integrating some information in a table (e.g. line 437-447, these questions could be indicated in a table to facilitate reading). Another important issue that should be addressed in this manuscript is the ethical issues in IBD applied omics research. This could provide a novel approach to its review.

Author Response

We appreciated the reviewer’s insightful comments.

We agree that various previous papers in IBD and omics exist, but none of them present the perspective offered in this review, i.e., the critical importance of omics integration as the only way to get a real grasp of IBD pathogenesis and develop novel therapeutic approaches based on hub target identification rather than the current one single pro-inflammatory molecular target approach. Current publications highlight the present technical capability of performing a multitude of omics and eventually correlating them with selected clinical parameters, but this only generates correlative descriptive data. Furthermore, the current IBD omics literature is spotty and selective but not comprehensive, and does not discuss why omic integration is essential and how it should be performed.

A graphical abstract has been submitted, but apparently not accessible to the reviewers.

As far the suggestion of integrating some information in a table rather than text form (e.g. line 437-447), we had originally considered this option, and this will be so done once the review process is completed and revisions allowed.

We believe that a discussion of potential ethical issues in IBD applied omics research is interesting but premature, considering that no literature is currently available on this topic.

Reviewer 3 Report

The text below contains comments on manuscript entitled “Omics and multi-omics in IBD: no integration, no break-throughs”

I think that a more profound argument should be presented why the focus on this review is the IBD.

To my opinion the manuscript is very well structured aiming to present in depth the novelties in the omics approaches concerning IBD. I think that at least two tables should be presented in the manuscript, which I think will significantly improve the manuscript. The first table could summarize the omics approaches and the main outcomes of them mentioned in section 5. The second table could present summary of the clinical approaches, the identified markers and response to therapies from section 6.

Minor editing of English language required

Author Response

We appreciated the reviewer’s insightful comments.

Why the focus of this review is IBD is clearly presented in the introduction: “This review will use inflammatory bowel disease (IBD) as a prototypical example of a complex clinical condition to illustrate and discuss multiple aspects of omics studies carried out in ulcerative colitis (UC) and Crohn’s disease (CD)”. While omics approaches can be applied to a number of other conditions the Special Issue to which the review has been submitted to is focused on the gastrointestinal tract: “The Role of Omics and Artificial Intelligence for the Personalized Management of Inflammatory and Neoplastic Diseases”.

We appreciate the suggestion of adding some tables. The submitted Table 1 does summarize the omics approaches relevant to IBD, but the main outcomes are too numerous to be listed in a table and their reliability, relevance and value are still in question as extensively discussed under Exposomics, Microbiomics, Immunomics, Epigenomics, Proteomics, metabolomics, lipidomics, and Single cell technologies, omics, multi-omics and spatial multi-omics. We feel very strongly that omics information is still partial and incomplete, and consequently too preliminary before assigning a final value to them. This also applies to the “identified markers” and “response to therapies” which have not been validated or reproduced. This lack of reliability of the single omics data so far reported in the IBD literature is one of the key messages of this review.

Reviewer 4 Report

The article "Omics and multi-omics in IBD: no integration, no breakthroughs" by Claudio Fiocchi is an interesting review on the fine distinction between expectations, marketing and real usefulness of Omics technologies in the understanding of IBD. Despite its sacriligeous tone, I think the review is spot-on and its main message (that lots of data do not translate into more knowledge) could very well be applied to many other molecular pathologies. This sentence in particular from the abstract rings extremely true "the majority of current reports are still based on simple associations of descriptive retrospective data from cross-sectional patient cohorts rather than more powerful longitudinally collected prospective datasets" and should be the starting point to start changing the current tide of articles. The review is overall very well written and I wholeheartily recommend its acceptance, pending some minor points (listed below).

- Paragraph 5.4 is called "epigenetics", breaking the succession of "-omics", while "epigenomics" exists and is a thriving field. The author should describe its main principles and go a bit more into the possibilites given by the recent epigenomics technologies into understanding the relationship between chromatine states and IBD. No mention to ATAC-Seq or even ChIP-Seq in a review of 2023 is a bit excessive.

- Connecting to the previous point, I believe the author is not delving deep enough into the many aspects, technologies and nuances associated to the "omics" revolution. All these techs are generally described in the "NGS" umbrella term. But there is no mention to RNA-Seq, ATAC-Seq, or even single-cell sequencing, which is allegedly revolutionary (if used well), especially in complex tissues (like the multihistological landscape provided by IBD)

- Line 600, while I concede that each omics database has its own structure, the author fails to mention that they essentially all use the same data standard, which is a great achievement (especially since both China and US use the same, despite the political tensions). FASTQ files follow the same standards in the Chinese National Genomics Data Center as well as in the NIH SRA database, or the European ENA collection. Same is true for other formats: BAM, VCF, TPM for gene expression. Also, the genome is generally the same (currently, the standard has been the genome version hg38). So I wouldn't be so pessimistic in a review already this provocative, in order not to give the impression that the situation is hopeless.

Author Response

We appreciated the reviewer’s positive and insightful comments. We are sorry that the reviewer perceived our tone as “sacrilegious”, which was not our intention, but rather that of trying to be realistic and provocative.

We apologize for the word “epigenetics” instead of “epigenomics”. It was an oversight that will be corrected once the review process is completed and revisions allowed.

We appreciate the comments on the lack of description of the multiple “aspects, technologies and nuances associated to the "omics" revolution”. This was done intentionally to avoid overburdening the primarily clinical readership of the review and not to assign ultimate determining values to each technology given the still evolving technological field.

The intention of mentioning some of the existing omics databases was just to provide some examples and highlight potential diversity among them, and we apologize if we gave a “hopeless” impression of the situation, which was not our intention. To correct this impression and avoid pessimism we included in 8.3. some of the informative comments provided by the reviewer: “On the other hand, while each omics database has its own structure, they essentially all use the same data standard. FASTQ files follow the same standards in the Chinese National Genomics Data Center as well as in the NIH SRA database, or the European ENA collection. The same is true for other formats: BAM, VCF, TPM for gene expression, and the genome is generally the same (the current standard is the genome version hg38).

Round 2

Reviewer 2 Report

Thanks to the authors for their corrections